# Sex differences in the prevalence of neural tube defects and preventive effects of folic acid (FA) supplementation among five counties in northern China: results from a population-based birth defect surveillance programme

Jufen Liu,[1,2] Jing Xie,[1,2] Zhiwen Li,[1,2] Nicholas D E Greene,[3] Aiguo Ren[1,2]

¹Institute of Reproductive and Child Health, Key Laboratory of Reproductive Health, National Health Commission of the People's Republic of China, Peking University, Beijing, China
²Department of Epidemiology and Biostatistics, School of Public Health, Peking University, Beijing, China
³UCL Great Ormond Street Institute of Child Health, University College London, London, UK

**Correspondence to**
Dr Zhiwen Li; lizw@bjmu.edu.cn

## ABSTRACT

**Objectives** Sex differences in prevalence of neural tube defects (NTDs) have previously been recognised; however, the different susceptibility of men and women have not been examined in relation to the effects of folic acid (FA) supplementation. We hypothesised that FA may have a disproportionate effect that alters the sex-specific prevalence of NTDs.

**Setting** Data from two time points, before (2003–2004) and after (2011–2016) the start of the supplementation programme, were obtained from a population-based birth defect surveillance programme among five counties in northern China. All live births (28 or more complete gestational weeks), all stillbirths of at least 20 weeks' gestational age and pregnancy terminations at any gestational age following the prenatal diagnosis of NTDs were included.

**Participants** A total of 25 249 and 83 996 births before and after the programme were included respectively.

**Primary and secondary outcome measures** The prevalence of NTDs by sex and subtype, Male:female rate ratios and their 95% CI were calculated.

**Results** Overall, NTDs were less prevalent among men than among women (rate ratio (RR) 0.92; 95% CI 0.90 to 0.94), so was anencephaly (RR 0.77; 95% CI 0.73 to 0.81) and encephalocele (RR 0.75; 95% CI 0.61 to 0.92), while spina bifida showed a male predominance (RR 1.10; 95% CI 1.05 to 1.15). The overall prevalence of NTDs decreased by 78/10 000 in men and 108.7/10 000 in women from 2003 to 2004 to 2011 to 2016. There was a significant sex difference in the magnitude of reduction, being greater in women than men, particularly for anencephaly.

**Conclusions** The prevalence of NTDs decreased in both sexes after the implementation of a massive FA supplementation programme. While female predominance was observed in open NTDs and total NTDs, they also had a greater rate of decrease in NTDs after the supplementation programme.

## Strengths and limitations of this study

► Our data are from a population-based surveillance system, which covered all live births (28 or more complete gestational weeks), and neural tube defect (NTD) cases from pregnancies of any gestational age.

► NTDs decreased among both men and women after the implementation of a massive folic acid (FA) supplementation programme in China, while these decreases were significantly greater in women than in men.

► The reduction in NTD cases during the post-FA period inversely mirrors the increase in plasma folate concentrations among pregnant women.

► Limitations of our study include that covering small regions and lacking of detailed subtypes of spina bifida.

## INTRODUCTION

The prevalence of many types of birth defects appears to differ by sex, with the majority of them being more prevalent among men,[1 2] including orofacial cleft (cleft lip with or without cleft palate), urinary system (hypospadias, hydronephrosis) and gastrointestinal tract (diaphragmatic hernia). However, some types of neural tube defects (NTDs) occur more frequently in women (eg, isolated cases of cranial defects, such as anencephaly).[3 4] Women tend to have anencephaly more often than men, while craniorachischisis and spina bifida involving the higher spine may also show a preponderance among women.[4] By contrast, there tends to be a higher proportion of men than women with spina bifida affecting the lower spine.[4] There are thought to be multiple initiation sites in human neural tube closure,[5] as described in other mammalian embryos. Neural tube closure occurs during weeks 3–4 of human gestation and the sex of the embryo

is differentially associated with a lack of closure of specific areas of the neural tube.[5]

Female predisposition to anencephaly is also observed among a number of mouse models,[3 6] but the mechanism underlying this has not been established. In mice, female embryos tend to be developmentally delayed compared with their male littermates at neurulation stages, leading to the hypothesis that the longer duration of cranial neural tube closure would lead to greater susceptibility to NTD-causing insult. However, going against this hypothesis, the delay in development may occur prior to neural tube closure, with female and male embryos taking the same time to complete cranial neurulation.[7] Other suggested mechanisms include sex differences in genetic risk, epigenetic factors, a differential rate of fetal loss and susceptibility to environmental influences.[8]

Among environmental factors, periconceptional supplementation with folic acid (FA) has been shown to dramatically reduce the risk for NTDs in offspring[9] and, conversely, suboptimal maternal folate status is a risk factor for NTDs.[10] Although a large number of studies have investigated the preventive effects of FA on NTDs, the influence of maternal FA supplementation on sex-associated risk for NTDs and their subtypes is unclear. Some studies have lacked data on the type of NTDs or sex,[11] while others have not found significant differences among subtypes of NTD.[12] In these studies, the apparent protective effects of FA do not seem to act preferentially in one region of the neural tube and there appears to be a general reduction in the occurrence of human NTDs.[13] In a study in Mexico in which weekly administration of 5 mg FA was associated with a 50% reduction in the incidence of NTDs, the main effect was on spina bifida, with a higher reduction in female cases.[8] In several countries, FA fortification (FAF) of food has been associated with a decreased frequency of spina bifida and anencephaly on a population level.[14 15] A recent study from South America showed that prevalence of NTDs, particularly anencephaly and cervicothoracic spina bifida, showed a greater reduction rate in women than in men after FAF resulting in a change of the sex ratio of infants with NTDs.[16]

In 2009, the Ministry of Health of China initiated a nationwide FA supplementation programme for the purpose of further decreasing the occurrence of NTDs in the country.[17] Women with a rural household registration and planning to become pregnant are eligible to obtain FA supplements free of charge through the maternal healthcare system. In a recent study, we showed that birth prevalence of NTDs in the five counties of Shanxi Province in northern China dropped dramatically, from 118.9/10 000 in 2000 to 31.5/10 000 in 2014.[18] In the present study, we examined the trends in NTD prevalence by sex in five counties of Shanxi Province, China, from 2003 to 2016, encompassing the introduction of a massive FA supplementation campaign. Characterisation of the sex distribution of NTDs may be helpful to better understand NTD development in association with FA supplementation.

## MATERIALS AND METHODS
### Birth defect surveillance
Five counties in Shanxi Province (Pingding, Xiyang, Taigu, Zezhou and Shouyang) were included in a population-based birth defects surveillance system in the current study. The surveillance system was similar to a previously reported system, created to meet the primary objective of monitoring the prevalence of major structural birth defects including NTDs.[19] A national FA supplementation programme was initiated in China in 2009; therefore, in the current study, we collected data at two time points: before (2003–2004) and after (2011–2016) the start of the supplementation programme. All pregnant women who resided in the study area for more than 1 year during the period were monitored. All live births (28 or more complete gestational weeks), all stillbirths of at least 20 weeks' gestational age and pregnancy terminations at any gestational age following the prenatal diagnosis of NTDs were included. Photographs of every suspected case were obtained. NTD diagnosis was conducted by local specialists in maternal-fetal medicine, and photographs were sent to a paediatrician at Peking University for confirmation. The study protocol was reviewed and approved.

### Statistical analysis
The prevalence of NTDs by infant sex was analysed separately for open NTDs (anencephaly, spina bifida)[20] and encephalocele. The denominator was the total number of all births and pregnancy that reached 28 or more complete gestational weeks by sex; the numerator was the number of NTD cases regardless of gestational age by sex. The prevalence of NTDs by sex and subtype were compared using $\chi^2$ tests. A difference in differences (DID) model was applied to compare sex differences between the time periods.[21] A two-tailed $p \leq 0.05$ was considered to indicate statistically significant differences. All statistical analyses were performed using the SPSS package V.20.0 (SPSS, Chicago, USA).

### Patient and public involvement
Our main analysis was based on the birth defects surveillance system; the information was anonymised. Patients and or public were not involved.

## RESULTS
### Prevalence before FA supplementation programme
The prevalence of NTDs by subtype and sex is shown in table 1. Before the programme, a total of 25 249 births and 341 cases of NTDs (156 men, 185 women) were recorded in the system. The overall prevalence of NTDs for both sexes was 135.1/10 000, comprising 118.6/10 000 among men and 152.9/10 000 among women. All subtypes except spina bifida were more prevalent among women with a male-to-female rate ratio (RR) of 0.60 (95% CI 0.45 to 0.80) and 0.79 (95% CI 0.77 to 0.81) for encephalocele and open NTDs, respectively (table 1).

**Table 1** Prevalence rates of NTDs (per 10 000 births) by type and sex in Shanxi Province, China, in 2003–2004 and 2011–2016

| NTDs type | Male | | | Female | | | M:F rate ratio | | |
|---|---|---|---|---|---|---|---|---|---|
| | Births | Cases | Rate | Births | Cases | Rate | RR | 95% CI | |
| **2003–2004 (Pre-FA supplementation)** | | | | | | | | | |
| | 13 150 | | | 12 099 | | | | | |
| Open NTDs | | 145 | 110.3 | | 168 | 138.9 | 0.79 | 0.77 | 0.81 |
| Anencephaly | | 65 | 49.4 | | 94 | 77.7 | 0.64 | 0.61 | 0.67 |
| Spina bifida | | 80 | 60.8 | | 74 | 61.2 | 0.99 | 0.94 | 1.04 |
| Encephalocele | | 11 | 8.4 | | 17 | 14.1 | 0.60 | 0.45 | 0.80 |
| Total NTDs | | 156 | 118.6 | | 185 | 152.9 | 0.78 | 0.76 | 0.80 |
| **2011–2016 (Post-FA supplementation)** | | | | | | | | | |
| | 43 538 | | | 40 458 | | | | | |
| Open NTDs | | 160 | 36.7 | | 158 | 39.1 | 0.94 | 0.92 | 0.96 |
| Anencephaly | | 64 | 14.7 | | 77 | 19.0 | 0.77 | 0.73 | 0.81 |
| Spina bifida | | 96 | 22.0 | | 81 | 20.0 | 1.10 | 1.05 | 1.15 |
| Encephalocele | | 17 | 3.9 | | 21 | 5.2 | 0.75 | 0.61 | 0.92 |
| Total NTDs | | 177 | 40.7 | | 179 | 44.2 | 0.92 | 0.90 | 0.94 |

NTDs, neural tube defects; RR, relative risk.

### Prevalence during FA supplementation programme

After the initiation of the programme, a total of 83 996 births and 356 cases of NTDs (177 men, 179 women) were recorded in the system. The overall prevalence for both sexes was 43.3/10 000, being 40.7/10 000 among men and 44.2/10 000 among women. Hence, in sum, NTDs were less prevalent among men than women (RR 0.92; 95% CI 0.90 to 0.94), as was anencephaly (RR 0.77; 95% CI 0.73 to 0.81) and encephalocele (RR 0.75; 95% CI 0.61 to 0.92), while spina bifida showed male predominance (RR 1.10; 95% CI 1.05 to 1.15) (table 1).

### Differences in changes in prevalence rates of NTDs

After the supplementation period, the prevalence of all types of NTDs was lower in both male and female fetuses. However, there was a significant sex difference in the magnitude of reduction, being more pronounced among women, especially for open NTDs including anencephaly (p<0.01). The overall prevalence of NTDs decreased by 78/10 000 in men and 108.7/10 000 in women in absolute terms; anencephaly decreased by 70.3% in men and 75.5% in women in relative term (table 2).

### DISCUSSION

This quasi-experimental population-based study examined whether introduction of population-wide FA supplementation was associated with changes in the sex ratio for NTDs prevalence in northern China, as a means to understand whether the preventive effects of FA supplementation differ by sex. We found that, after the introduction of the national FA supplementation campaign, the prevalence of total NTDs and open NTDs (including anencephaly and spina bifida) decreased among both sexes. However, the magnitude of decline was significantly greater among women than men.

**Table 2** Differences in changes in prevalence rates of NTDs (per 10 000 births) by type and sex in Shanxi Province, China, in 2003–2004 and 2011–2016

| Types | Male rate | | Female rate | | Female–Male | |
|---|---|---|---|---|---|---|
| | Absolute | Relative | Absolute | Relative | DID value | P values |
| Open NTDs | 73.5 | 66.7 | 99.8 | 71.9 | 26.3 | 0.001 |
| Anencephaly | 34.7 | 70.3 | 58.7 | 75.5 | 23.9 | 0.001 |
| Spina bifida | 38.8 | 63.8 | 41.1 | 67.3 | 2.4 | 0.765 |
| Encephalocele | 4.5 | 53.3 | 8.9 | 63.1 | 4.4 | 0.213 |
| Total NTDs | 78.0 | 65.7 | 108.7 | 71.1 | 30.7 | 0.007 |

DID, difference in difference; NTDs, neural tube defects.
Absolute rate of decrease in prevalence=prevalence of group before folic acid supplementation–prevalence of group after folic acid supplementation.
Relative rate of decrease in prevalence=absolute rate of decrease in prevalence/prevalence of group before folic acid supplementation.

The prevalence of NTDs in Shanxi Province in northern China is historically high, being 105.5 per 10 000 births in the late 1980s[22] and 138 per 10 000 births in 2003.[23] In 2009, the Ministry of Health of China initiated a nationwide FA supplementation programme for the purpose of further decreasing the occurrence of NTDs in the country.[17] Women with a rural household registration and planning to become pregnant are eligible to obtain FA supplements free of charge through the maternal healthcare system. In a recent study, we showed that birth prevalence of NTDs in the five counties of Shanxi Province in northern China dropped dramatically, from 118.9/10 000 in 2000 to 31.5/10 000 in 2014.[18] However, differences between the sexes among this population have rarely been examined. One previous study reported a female predominance among anencephaly cases (male-to-female RR 0.49; 95% CI 0.30–0.79) but not among spina bifida (RR 0.90; 95% CI 0.55 to 1.45) or encephalocele (RR 1.03; 95% CI 0.40 to 2.69) cases in 2006[23]; however, that study was conducted before the national FA supplementation programme and how the NTD prevalence has changed in each sex in association with the programme remains unknown.

Since 2009, the nationwide FA supplementation programme has provided FA supplements, free of charge, to all women who have a rural registration and who plan to become pregnant. As we previously reported, the prevalence of NTDs was dramatically lower in 2012–2014 than in 2009 in Shanxi Province. This decrease may be partially attributable to the programme; however, this decreasing trend has not continued since 2014. In fact, the prevalence in the same area was higher in 2015–2016 (32.8 per 10 000 births) than in 2014 (31.5 per 10 000 births). Moreover, the prevalence of NTDs in this area remains high when compared with the rates in the USA,[15] Canada,[24] Germany,[25] England, Wales[26] and Western Australia.[27] The compliance of women taking the FA pills may be attributable to the high prevalence. A previous study showed the percentage of folic acid supplementation before the last menstrual period increased only from 30% to 43% and the supplementation adherence of ≥8 days/10 days showed a decrease.[28] Other reasons may need further study.

In addition to the NTD prevalence data, our previous cross-sectional surveys on the use of FA and blood folate concentration among pregnant women provide external evidence to test the correlation between folate status and NTDs. Those two surveys took place in the same area, one before the FA programme started (2003–2004)[29] and the other after it started (2011–2012).[30] Detailed descriptions of the two studies have already been published.[30 31] Briefly, a face-to-face interview with a structured questionnaire including information on FA supplementation was conducted among women who was in their late first trimester or early second trimester of pregnancy. A 5 mL non-fasting blood sample was collected at the time of recruitment and plasma levels of folate were determined using a microbiological assay.[32]

Our previous cross-sectional study showed that the proportion of people taking FA supplements increased from 9.2% in 2003–2004[33] to 66.3% in 2011–2012[30]. Comparison of population data on blood folate concentration among pregnant women and NTD prevalence data indicated a clear trend: plasma folate concentration increased from 10.4 nmol/L in 2003–2004[33] to 33.4 nmol/L in 2011–2012[22] while the prevalence of NTDs decreased from 117.8/10 000 births to 60.3/10 000 births, respectively.[19] Women who took FA supplements had significantly higher folate concentrations (median 41.9 nmol/L) than those who did not (13.2 nmol/L) in 2003–2004. In contrast, these values were 50.7 nmol/L and 17.3 nmol/L, respectively, in 2011–2012. The prevalence of NTDs was lower among both men and women after the programme. However, the effect was not the same for men and women, being greater among women than men. For open NTDs, the decrease was 26.3/10 000 higher among women than men; for total NTDs (including encephalocele), the decrease was 30.7/10 000 higher among women than men in the current study.

Studies on mice have indicated that female embryos are more susceptible to NTDs than males in some, but not all, genetic models.[34] For example, there is a higher rate of cranial NTDs in mice carrying mutations in *Trp53*,[6] *Pax3*,or *Nf1*[35] but not *Gldc*.[36] Sex differences in NTD susceptibility in *Trp53* null mice may result from the presence of two X chromosomes and independent of the Y chromosome.[6] Given the association between abnormal one-carbon metabolism or impaired methylation and NTDs, it has been hypothesised that the higher rate of cranial NTDs in females may be an epigenetic phenomenon. The demand for genomic DNA methylation is higher in females, which methylate most of the DNA in the large inactive X chromosome after every cell division. It is speculated that this may reduce the methylation capacity available for other key cellular needs.[3] In the splotch ($Sp^{2H}$; *Pax3* mutant) strain, female susceptibility to exencephaly is exacerbated by dietary folate deficiency to a greater extent than in males.[37] While the aetiology of human NTDs is more complex than in single-gene models, if such a scenario occurs in some human NTDs, then it might be predicted that an overall increase in population-level folate status may have a greater effect on NTDs in women than in men. This is consistent with our findings for anencephaly rates.

Genetic and environmental factors may also interact to determine murine sex ratios, with female embryos being more susceptible to lethality. In the splotch strain, for example, arsenite treatment and advanced maternal age are both associated with higher rates of in utero death and NTDs.[38] Both factors result in elevated sex ratios (male:female) among litters, which suggests increased loss of female embryos by resorption. This effect is genotype-dependent such that a greater effect, with lower sex ratio, is noted as the number of *Sp*(*Pax3* mutant) alleles increases.

A study of human NTDs indicated an overall excess of affected women, with the overall male:female sex distribution of unaffected carrier or transmitting individuals being 0.64, and an even greater excess of female gene carriers, with a male:female ratio of 0.52, when only closely related relatives are counted.[39] To fully understand the genetic influences of NTDs, future studies are needed to investigate the importance of epigenetic factors and imprinting effects, methylation status and folate supplementation, as well as the gender of the child in altering NTD risk.

In fact, the larger decline in NTD prevalence among women is somewhat expected given that it was higher in the presupplementation time period. The difference between sexes which is less pronounced is consistent with prevalences beginning to converge toward the prevalence of NTDs that are not preventable by FA.

Our data are from a population-based surveillance system, which covered all live births (28 or more complete gestational weeks), and NTD cases from pregnancies of any gestational age. This could give a more precise estimation of NTD rate than hospital-based surveillance systems that include only NTD cases in pregnancies of 28 gestational weeks or greater in China.[40]

Three main limitations of our study should be mentioned. First, we included data from only five counties, and the results may not be generalisable to the entire province or country. Second, we did not include the detailed subtypes of spina bifida, such as high-level and low-level lesions, which may show different patterns. A future study may be useful to address this question. Third, as an ecological design in which some confounding factors might affect our results, the massive FA supplementation programme was the only major factor that may affect the population rate of NTDs in the study. Improvement in living standards and prenatal diagnosis may also impact the rate of NTDs, but these potential factors are not predicted to have a differential effect on men and women.

In conclusion, the prevalence of NTDs decreased in both sexes after the implementation of a massive FA supplementation programme, but to a greater degree (both total NTDs and open NTDs) in women.

**Correction notice** Since this paper was first published online, the OA licence has changed from CC-BY-NC to CC-BY.

**Contributors** ZL designed research; JL and ZL conducted research; JL analysed the data and wrote the first draft of the manuscript; JX collected the data and analysed the data, AR supervised data collection and revised the draft; and NDEG revised the draft and provided critical suggestions on the study plan. All authors read, reviewed and approved the final manuscript.

**Funding** This work was supported in part by the National Key Research and Development Program, Ministry of Science and Technology, P.R. China (grant no. 2016YFC1000500), National Natural Science Foundation of China (no. 81373014, no. 81511130088 and no. 81202265) and Medical Research Council (UK, N003713). NDEG is supported by Great Ormond Street Hospital Children's Charity.

**Competing interests** None declared.

**Patient consent** Not required.

**Ethics approval** The Institutional Review Board of Peking University.

**Provenance and peer review** Not commissioned; externally peer reviewed.

**Data sharing statement** ZL took full responsibility to the data and data could be shared upon requirement.

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
