## [Reviewer comments · BMJ Open]

ARTICLE DETAILS

TITLE (PROVISIONAL)	Sex differences in the prevalence of neural tube defects and preventive effects of folic acid (FA) supplementation among five counties in northern China: Results from a population-based birth defect surveillance program
AUTHORS	Liu, Jufen; Xie, Jing; Li, Zhiwen; Greene, ND; Ren, Aiguo

VERSION 1 – REVIEW

REVIEWER	Sarah Tinker Centers for Disease Control and Prevention, USA The opinions and views in this review are those of the reviewer and do not represent the views of the CDC
REVIEW RETURNED	22-Mar-2018

GENERAL COMMENTS	The manuscript presents important population-based data on the prevalence of neural tube defects (NTDs) before and after a population-level supplementation program in 5 provinces of China. I think it is interesting to look at sex differences in prevalence rates, but the ultimate importance of these potential differences in clinical or public health practice are unclear. The analysis itself is done well, but I disagree with one of the major interpretations (described in more detail below). In addition, I worry that selective terminations based on sex may be biasing all of the results, given results from other studies that do not find a sex difference in the prevalence of these defects. General comments: - Percentage decreases are a function of baseline prevalence. The larger percentage decrease among females appears to be largely a function of their higher prevalence prior to the FA supplementation program. The prevalence differences between the sexes are much smaller – absolutely and proportionally – after the supplementation program compared to before the supplementation program. If the prevalence of NTDs was truly higher among females prior to the supplementation program (which I question – see next point), and folic acid worked EQUALLY well in preventing NTDs among both sexes, we would expect to see a larger percentage decrease among females as the prevalence of both sexes becomes closer to the prevalence of NTDs that cannot be prevented by folic acid.- I am concerned that the higher prevalence among females, before and after the supplementation program, may be attributable to selective termination of female fetuses prior to 28 weeks. In the numerator of the
--

	prevalence estimates are all cases of NTDs, regardless of gestational age. But a fetus without an NTD had to survive to 28 weeks to be counted in the denominator. Selective termination of females over males would lead to a smaller denominator, while the number of NTD cases would not be affected. Sex differences for NTDs were not reported in a recent population-based study of pregnancies in several areas of the U.S. (please see Am J Med Genet Part A 167A:1071–1081).  - The data presented on blood folate levels and NTD prevalence appear to have been previously published and seem out of place in the Methods and Results; they should be presented only in the Discussion. - There is an error in Table 1. Specific comments:  - Not all of the information in the “Strengths and limitations of this study” describes strengths or limitations; some of the bullets are just summaries. - Introduction  o Reference 1 is over 45 years old. Have these patterns been seen in more recent data? o The authors list “defects of the sex organs” as one of the types of birth defects that appear to differ by sex. Wouldn’t these defects differ by sex by definition? o Reference 2 is titled as a hypothesis. Is there population-based data supporting this statement? And in this sentence, what exactly do the authors mean by “cranial defects”? o Paragraph 1, Sentence 4: A reference is needed for the statement that the sex of the embryo is differentially associated with lack of closure of specific areas of the neural tube. o Paragraph 1, Sentence 5: A reference is also needed for the description of the differences by sex in this sentence. o Paragraph 2, sentence 1: Need reference for statement about folate status and risk for NTDs (see Crider et al. BMJ). o Page 5, first sentence: By generalized, do the authors mean across sex and subtype? o Paragraph 2, last sentence: This statement is inaccurate. More recent estimates show a decrease for both anencephaly and spina bifida attributable to folic acid fortification in the U.S. (please see Williams et al. MMWR 2015;64(1):1 PubMed -5) o Paragraph 3, sentence 2: What led to the initial decrease? o Paragraph 4: How would sex differences in the prevalence of NTDs or the impact of folic acid on the prevention of NTDs impact prevention efforts? NTDs occur before the sex of the baby is known (except in some cases of assisted reproductive technology). Folic acid is preventive regardless of sex, even if magnitudes differ. - Methods
--	--

	 o Why were only two years of pre-supplementation program data used (2003-2004), while 7 years of post-supplementation program data were used? o The ascertainment of NTDs appears to be comprehensive by including all pregnancy outcomes at any gestational age. In addition, the case classification described by the authors is excellent and greatly reduces the likelihood of outcome misclassification. Many studies of NTDs fail to include encephalocele, an important contributor to overall NTD prevalence, which these authors included. Well done! o In the first sentence of the “Statistical analysis” section, the authors should make clear that while they consider anencephaly and spina bifida collectively as “open NTDs”, they also consider them separately. o The description of the denominator needs to specify that it is all births and pregnancies that complete 28 weeks BY SEX. - Results  o The authors need to provide the sex-specific denominators. o Paragraph 1, last sentence: This statement isn’t correct; the ratio is approximately equal for spina bifida. o As mentioned above, the entire section “NTD Prevalence in Association with Blood Folate” presents no new data and doesn’t seem related to this analysis. It should be removed and discussed only in the Discussion. - Discussion  o I’m not sure it’s appropriate to call the study design “quasi-experimental”. While the supplementation programs were implemented on a population level, individuals still had the option to not take the supplements (unlike fortification which is truly experienced by everyone in the population). There is still the potential for the ecologic fallacy given that we don’t know (from these data) that the women who took the supplements had reduced risk. o Paragraph 2: Unless something about the supplementation program is changing, a continued decrease in NTD prevalence would not be expected. In the U.S. the large decline attributable to fortification was seen over the few years that it was implemented and then the rate has remained relatively stable since. o There are many reasons why prevalence estimates might differ from those in other countries. Did all of the cited studies include all birth outcomes, as this study did? If not, they will have lower prevalence estimates. How does the distribution of RBC folate in the study population compare to the distribution in the populations of these other studies. o Paragraph 5: This paragraph is only about mice. The authors need to discuss human data (which are available in the published literature). - Table 1
--	---

	 o There is an error in the Pre-FA supplementation section for the RR. The RRs that are presented are in the wrong order; starting with anencephaly the correct number is actually the one in the row below it. o The significance of the bolding is unclear and it is distracting. o While the percentage decrease is higher for females, as described by the authors, another important result is that the percentage DIFFERENCE between males and females also becomes much smaller in the post-FA supplementation period.  - Table 2  o These data do not add any new information. I suggest providing estimates of the percentage difference between sexes here instead. - Figures 1 and 2  o These figures are bleary and small and, as suggested earlier, should be omitted based on the fact that these data are not from this study.
--	--

REVIEWER	Somchit Jaruratanasirikul Prince of Songkla University
REVIEW RETURNED	02-Apr-2018

GENERAL COMMENTS	This is a population-based study of the prevalence of neural tube defects (NTD) in Shanxi County, Northern China, comparing between before folic acid supplementation (2003-2004) and after folic acid supplementation (2011-2016). The results of the study showed a significantly higher prevalence of NTD in females than in males in the time period before folic acid supplementation and a significant overall decrease of the prevalence of NTD after folic acid supplementation, particularly in females. Comments  1. The Introduction section has 2 pages which is too long. Some parts in paragraphs 2 and 3 can be moved to the Discussion section. 2. In the Abstract, the authors stated that “few population-based studies have examined sex difference among infants with NTD”. Actually, sex is one of many parameters that are used for comparison in most of the previous studies of the prevalence of NTD; some without sex differences and some with sex differences. 3. The strength of this study is the measurement of blood folic acid levels. However, the authors should give more details of the blood collection method and the method of folic acid measurement, not just the microbiological assay and a reference. Also, the interassay and intra-assay coefficients of variations should be stated in the method of measurement. 4. The authors showed the decreased prevalence of NTD and the increased level of folic acid before and after the folic acid supplementation using a histogram. It may be better if the authors showed these results using correlation statistics. 5. Figure 2 shows the blood levels of folic acid comparing participants who took folic acid and those who did not take folic acid, which was already described in the Results section. Hence, Figure 2 should be deleted.
--

REVIEWER	Saeed Dastgiri Tabriz University of Medical Sciences Tabriz, Post Code: 5166615739, Iran
REVIEW RETURNED	07-Apr-2018

GENERAL COMMENTS	This is a study to investigate the association of folic acid supplementation and NTDs (by sex) in northern China. The study is well organized/written. Some points/comments: (1) How the folic acid supplementation in mothers was determined during the study period? The coverage rate? The accuracy of data? More details are needed. (2) The details of the basic characteristics of mothers and neonates are needed (as a separate table). (3) This is an ecological design in which many surrounding confounding factors might usually affect the data/results. What are the confounders in this study? How investigators have controlled the role of those confounders? (4) The role of the improvements in the diagnosis and surveillance of NTDs over the study period in detection of NTD cases need to be indicated/addressed.
--

REVIEWER	Vijaya Kancherla Emory University, USA
REVIEW RETURNED	11-Apr-2018

GENERAL COMMENTS	This is an important work and a well-written paper. The authors have very clearly presented the rationale and findings on the impact of folic acid supplementation on overall reduction in NTDs, and sex-specific reductions. I have a few comments for minor revisions and to require some clarifications. 1. There are a few acronyms in the abstract and manuscript that need to be defined at their first use. (e.g., NTD) 2. Refer to studies from the US and other countries with active population-based birth defects surveillance registries to comment on sex difference in prevalence of NTDs (overall and by NTD subtype) during pre- and post-mandatory folic acid fortification periods. There is a reference from Mexico; however, it would be of interest to see what was observed in other countries that have data to compare with the current study. 3. In the Methods section, clarify how sex-specific prevalence was estimated. Who were included in the denominators? It would also be informative if sex-specific prevalence or proportion of cases were presented by pregnancy outcome (live birth / still birth / elective termination) before and after fortification either in the main manuscript or as a supplemental table. 4. On page 9, first para, authors say that the decrease in prevalence of NTD in China post supplementation is still higher than USA, Canada, Germany, England, and Western Australia. US, Canada and Western Australia have mandatory fortification with folic acid.
--

	Would it be possible that the supplementation program in China is still dependent on compliance of women taking the folic acid pills? That would be different in countries with mandatory fortification, as there is no reliance on personal behaviors to take supplements. Are there any data from China on compliance to folic acid pill intake in the target population? If yes, perhaps you can include it in your discussion to discuss why prevalence in China is higher in spite of the overall drop from the pre-supplementation period. 5. In Table 1, add columns with total births (indicating denominator values) for males and females (i.e., next to frequency of cases for each sex group) 6. Was unable to read the figures clearly the way it was printed in the PDF. Can increase the size and resolution of the figures.
--	--

VERSION 1 – AUTHOR RESPONSE

Reviewer: 1

Reviewer Name: Sarah Tinker

Institution and Country: Centers for Disease Control and Prevention, USA. The opinions and views in this review are those of the reviewer and do not represent the views of the CDC Please state any competing interests or state 'None declared': None declared

Please leave your comments for the authors below Please see attached.

3/22/2018

Peer Review

BMJ Open Manuscript bmjopen-2018-022565 “Sex differences in the prevalence of neural tube defects and preventive effects of folic acid (FA) supplementation in northern China: Results from pre- and post-FA supplementation”

The manuscript presents important population-based data on the prevalence of neural tube defects (NTDs) before and after a population-level supplementation program in 5 provinces of China. I think it is interesting to look at sex differences in prevalence rates, but the ultimate importance of these potential differences in clinical or public health practice are unclear. The analysis itself is done well, but I disagree with one of the major interpretations (described in more detail below). In addition, I worry that selective terminations based on sex may be biasing all of the results, given results from other studies that do not find a sex difference in the prevalence of these defects.

Thank you for recognizing the significance of our study. Our data were obtained from a population-based birth defect surveillance program among five counties in northern China, which is the region with highest prevalence of NTDs in China.

The importance of the current study was to discover sex-specificity of prevalence rates and the preventive effect of folic acid on NTDs by sex, which could help reveal the mechanisms underlying etiology and prevention of NTDs.

The reviewer makes an important point about possible bias in countries where selective termination of pregnancy. In China, there are strict "Regulations on the Prohibition of the Termination of Pregnancy for Non-medical Purposes and Sex Identification and Sex Selection of Fetus by Using Ultrasound Machine" and violation of the regulations will lead to imposition of a fine and administrative penalty. Hence, we do not think selective termination would influence the results in this study region (*Please*

see: *Sex identification of fetus banned by law. China Popul Today. 1998 Dec;15(5-6):4.*). In addition, an empirical study revealed that fetal sex was not associated with termination of pregnancy at <28 gestational weeks (*Please see Zheng X, Pei L, Chen G, Song X, Wu J, Ji Y. Periconceptional Multivitamin Supplementation Containing Folic Acid and Sex Ratio at Birth in a Chinese Population: a Prospective Cohort Study. Paediatr Perinat Epidemiol. 2015 Jul;29(4):299-306.*).

General comments:

1. Percentage decreases are a function of baseline prevalence. The larger percentage decrease among females appears to be largely a function of their higher prevalence prior to the FA supplementation program. The prevalence differences between the sexes are much smaller – absolutely and proportionally – after the supplementation program compared to before the supplementation program.
Yes, because the prevalence of both males and females decreased after the supplementation program, while female shares a higher prevalence before the program. For this reason we considered the difference of the change of prevalence to compare sex differences during the pre- and after the FA supplementation program. A difference in differences (DID) model was applied to reflect the true difference between pre- and after-the program. The same method has been used in South America (*Please see Poletta FA, Rittler M, Saleme C, et al, Neural tube defects: Sex ratio changes after fortification with folic acid. PLoS One. 2018 Mar 14;13(3):e0193127. doi: 10.1371/journal.pone.0193127. eCollection 2018.*).
2. If the prevalence of NTDs was truly higher among females prior to the supplementation program (which I question – see next point), and folic acid worked EQUALLY well in preventing NTDs among both sexes, we would expect to see a larger percentage decrease among females as the prevalence of both sexes becomes closer to the prevalence of NTDs that cannot be prevented by folic acid.

This is an interesting point:

Firstly, we do believe that the data showing higher prevalence of NTDs among females pre-supplementation is a 'real' finding. This was not a surprising observation, being not only observed in our study, but also in other published studies (*Please see [1]Tennant PW, Samarasekera SD, Pless-Mullooli T, et al. (2011) Sex differences in the prevalence of congenital anomalies: a population-based study. Birth Defects Res A Clin Mol Teratol 91,894-901; [2]Juriloff DM, Harris MJ. Hypothesis: the female excess in cranial neural tube defects reflects an epigenetic drag of the inactivating X chromosome on the molecular mechanisms of neural fold elevation. Birth Defects Res A Clin Mol Teratol. 2012; 94:849-855.*).

After the implementation of supplementation, the prevalence of NTDs among both males and females decreased. However, the prevalence of NTDs among females was still higher than males - that's why we consider the difference of the change of prevalence to compare sex differences during the pre- and after the FA supplementation program. Our result showed that the prevention was not equal to both sexes. There was a greater effect in females, which is in accordance with the reviewer's suggestion that the prevalence in both sexes may becoming closer to the rate that is folic acid-resistant.

The same method has been used in South America (*Please see Poletta FA, Rittler M, Saleme C, et al, Neural tube defects: Sex ratio changes after fortification with folic acid. PLoS One. 2018 Mar 14;13(3):e0193127. doi: 10.1371/journal.pone.0193127. eCollection 2018.*). The difference between the study of South America and our study was, South America study evaluated the effect of fortification while we studied the effect of folic acid supplementation. While there is no folic acid fortification in China, and we still observed the trend, which is the significant meaning of our study.

3. I am concerned that the higher prevalence among females, before and after the supplementation program, may be attributable to selective termination of female fetuses prior to 28 weeks. In the numerator of the prevalence estimates are all cases of NTDs, regardless of gestational age. But a fetus without an NTD had to survive to 28 weeks to be counted in the denominator. Selective termination of females over males would lead to a smaller denominator, while the number of NTD cases would not be affected. Sex differences for NTDs were not reported in a recent population-based study of pregnancies in several areas of the U.S. (please see *Am J Med Genet Part A* 167A:1071–1081).
(Please see response above)
4. The data presented on blood folate levels and NTD prevalence appear to have been previously published and seem out of place in the Methods and Results; they should be presented only in the Discussion.
We agree, we have moved this section to Discussion.
5. There is an error in Table 1.
We had corrected the data in the current manuscript.

Specific comments:

Not all of the information in the “Strengths and limitations of this study” describes strengths or limitations; some of the bullets are just summaries.

We revised the section of “Strengths and limitations of this study” and copied as following:

- I Neural tube defects decreased significantly among both males and females after the implementation of a massive folic acid supplementation program in China.
- I These decreases were significantly greater in females than in males.
- I The reduction in NTD cases during the post-FA period inversely mirrors the increase in plasma folate concentrations among pregnant women.

Introduction

1. Reference 1 is over 45 years old. Have these patterns been seen in more recent data?
Although the pattern of sex difference was firstly observed in the early 1970s, it is still present as a recent study reported. (Please see [1]Diogenes TCP, Mourato FA, de Lima Filho JL, Mattos SDS. *Gender differences in the prevalence of congenital heart disease in Down's syndrome: a brief meta-analysis BMC Med Genet.* 2017 Oct 6;18(1):111. doi: 10.1186/s12881-017-0475-7.[2]Tennant PW, Samarasekera SD, Pless-Mulloli T, et al. (2011) *Sex differences in the prevalence of congenital anomalies: a population-based study. Birth Defects Res A Clin Mol Teratol* 91,894-901)
2. The authors list “defects of the sex organs” as one of the types of birth defects that appear to differ by sex. Wouldn't these defects differ by sex by definition?
We may not clearly express our meaning. Here we mean that defects of the sex organs was more prevalent in males, for example, Hypospadias. We revised this sentence and copied as following: “The prevalence of many types of birth defects appear to differ by sex, with the majority of them being more prevalent among males, such as orofacial cleft (cleft lip with or without cleft palate), urinary system (hypospadias, hydronephrosis), and gastrointestinal tract (diaphragmatic hernia).”
3. Reference 2 is titled as a hypothesis. Is there population-based data supporting this statement? And in this sentence, what exactly do the authors mean by “cranial defects”?
Reference 2 reviews the possible mechanism underlying female predisposition to NTDs in both human and mouse (Please see Juriloff DM, Harris MJ. *Hypothesis: the female excess in cranial neural tube defects reflects an epigenetic drag of the inactivating X chromosome on the molecular mechanisms of neural fold elevation. Birth Defects Res A Clin Mol Teratol.* 2012; 94:849-855).

Cranial NTDs refers to anencephaly in humans and exencephaly in mouse (the developmental forerunner of anencephaly).

4. Paragraph 1, Sentence 4: A reference is needed for the statement that the sex of the embryo is differentially associated with lack of closure of specific areas of the neural tube.
Reference 2 is added in sentence 4 (*Please see Juriloff DM, Harris MJ. Hypothesis: the female excess in cranial neural tube defects reflects an epigenetic drag of the inactivating X chromosome on the molecular mechanisms of neural fold elevation. Birth Defects Res A Clin Mol Teratol. 2012; 94:849-855*).
5. Paragraph 1, Sentence 5: A reference is also needed for the description of the differences by sex in this sentence.
Reference 4 is added in sentence 3 (As we revised the introduction, sentence 5 is sentence 3 now) (*Please see Tennant PW, Samarasekera SD, Pless-Mulloli T, et al. Sex differences in the prevalence of congenital anomalies: a population-based study. Birth defects research Part A, Clinical and molecular teratology 2011;91(10):894-901*).
6. Paragraph 2, sentence 1: Need reference for statement about folate status and risk for NTDs (see Crider et al. BMJ).
Reference has been added in sentence 1, Paragraph 2.
7. Page 5, first sentence: By generalized, do the authors mean across sex and subtype?
It means that folate was generally involved in prevention NTDs. We rephrased this sentence in the revision and copied as following: "In these studies the apparent protective effects of FA does not seem to act preferentially in one region of the neural tube and there appears to be a general reduction in the occurrence of human NTDs."
8. Paragraph 2, last sentence: This statement is inaccurate. More recent estimates show a decrease for both anencephaly and spina bifida attributable to folic acid fortification in the U.S. (please see Williams et al. MMWR 2015;64(1):1 PubMed -5)

Thanks for the careful and constructive suggestion, the reference we cited was study in 2002 and we updated the reference (Williams J, et al. MMWR Morb Mortal Wkly Rep 2015;64(1):1-5). The revision was copied as following: "Some studies have found that FA fortification of food is associated with a decreased frequency of spina bifida and anencephaly on a population level"

9. Paragraph 3, sentence 2: What led to the initial decrease?
The prevalence of NTDs was 138 per 10,000 births in 2003 while it's 105.5 per 10,000 births in the late 1980s. There was no decrease before folic acid supplementation.
10. Paragraph 4: How would sex differences in the prevalence of NTDs or the impact of folic acid on the prevention of NTDs impact prevention efforts? NTDs occur before the sex of the baby is known (except in some cases of assisted reproductive technology). Folic acid is preventive regardless of sex, even if magnitudes differ.
We agree that FA is preventive in both sexes – this was an important finding of the study. In considering potential adjunct therapy it is therefore important to consider that FA-resistant NTDs occur in both sexes.

Methods

1. Why were only two years of pre-supplementation program data used (2003-2004), while 7 years of post-supplementation program data were used?
The reason we use the data of two time points was that we hope to compare the NTDs prevalence and other indicators, such as use of FA and blood folate concentration, before and after the supplementation program. For the period before the supplementation program, we have cross-sectional survey data in 2003–2004, so we used these two years data; for the period after the

supplementation, we'd wanted to examine the recent trend, so we used data for a long period (2011–2016), for which we had a similar cross-sectional survey at that time.

2. The ascertainment of NTDs appears to be comprehensive by including all pregnancy outcomes at any gestational age. In addition, the case classification described by the authors is excellent and greatly reduces the likelihood of outcome misclassification. Many studies of NTDs fail to include encephalocele, an important contributor to overall NTD prevalence, which these authors included. Well done!

Thanks. As we had long history to study NTDs in China, the birth defect surveillance system in our research field was rooted well.

3. In the first sentence of the “Statistical analysis” section, the authors should make clear that while they consider anencephaly and spina bifida collectively as “open NTDs”, they also consider them separately.

According to the practical clinical classification of spinal neural tube defects (See McComb JG, Childs Nerv Syst. 2015;31(10):1641-57), NTD was broadly divided into closed NTD and open NTD. We add the reference in current version.

Anencephaly and spina bifida were referred as open NTDs in our study. To clearly reveal the trends and difference of NTDs and its subtype, we studied the subtype separately and also combined them in total. The reason for consideration of the subtypes of open NTDs separately was the prior evidence for sex differences in anencephaly particularly.

4. The description of the denominator needs to specify that it is all births and pregnancies that complete 28 weeks BY SEX.

Thanks for the suggestion. We added the description by sex and we copied as following “The denominator was the total number of all births and pregnancy that reached 28 or more complete gestational weeks by sex; the numerator was the number of NTDs cases regardless of gestational age by sex.”

Results

1. The authors need to provide the sex-specific denominators.

Thanks for the suggestion. We added all the births by sex as sex-specific denominators.

2. Paragraph 1, last sentence: This statement isn't correct; the ratio is approximately equal for spina bifida.

Thanks for the suggestion. We revised the sentence as following “All subtypes except spina bifida were more prevalent among females”.

3. As mentioned above, the entire section “NTD Prevalence in Association with Blood Folate” presents no new data and doesn't seem related to this analysis. It should be removed and discussed only in the Discussion.

We moved this section to paragraph 4 in Discussion part.

Discussion

1. I'm not sure it's appropriate to call the study design "quasi-experimental". While the supplementation programs were implemented on a population level, individuals still had the

option to not take the supplements (unlike fortification which is truly experienced by everyone in the population). There is still the potential for the ecologic fallacy given that we don't know (from these data) that the women who took the supplements had reduced risk.

The quasi-experimental study here means that we had the data and survey before and after the supplementation which provide opportunity to compare the changes. When the first survey was conducted, we did not know about the subsequent massive folic acid supplementation campaign for all childbearing age women.

For sure, the reviewer's comments are right. Behavior of individuals who take or don't take the supplements and their outcome still needs further study which will be based on more comprehensive data collection and analysis.

2. Paragraph 2: Unless something about the supplementation program is changing, a continued decrease in NTD prevalence would not be expected. In the U.S. the large decline attributable to fortification was seen over the few years that it was implemented and then the rate has remained relatively stable since.

Yes, we quite agree the reviewer's opinion. The decreasing trend has not continued since 2014 in our study.

3. There are many reasons why prevalence estimates might differ from those in other countries. Did all of the cited studies include all birth outcomes, as this study did? If not, they will have lower prevalence estimates. How does the distribution of RBC folate in the study population compare to the distribution in the populations of these other studies.

Yes, the reasons of different NTDs prevalence among countries are likely to involve multiple factors. We did not include the analysis of the reasons as it's not the emphasis of our study. To be more precise, we added one sentence that the reasons still need further study. However, the RBC folate concentration was not available for all the studies. The comparison of NTDs prevalence of different countries may be another study to be stressed.

4. Paragraph 5: This paragraph is only about mice. The authors need to discuss human data (which are available in the published literature).

Thanks for the suggestion, we added the discussion on human data in this paragraph.

Table 1

1. There is an error in the Pre-FA supplementation section for the RR. The RRs that are presented are in the wrong order; starting with anencephaly the correct number is actually the one in the row below it.

Thanks for your careful check and the data was skipped one line. We had corrected the data in the current manuscript.

2. The significance of the bolding is unclear and it is distracting. We remove the bolding fonts all through the table 1.

3. While the percentage decrease is higher for females, as described by the authors, another important result is that the percentage DIFFERENCE between males and females also becomes much smaller in the post-FA supplementation period.

Yes, we quite agree the reviewer. We add this finding in the revision.

- Table 2

These data do not add any new information. I suggest providing estimates of the percentage difference between sexes here instead.

Thanks for the suggestion. We add the percentage of the change in Table 1. However, the difference in differences (DID) model we used currently was to calculate the absolute change between the time periods, so we still keep the absolute percentage (Please see Ashenfelter O et al, 1985;67(4):648-60).

- Figures 1 and 2

These figures are bleary and small and, as suggested earlier, should be omitted based on the fact that these data are not from this study.

Yes, we deleted Figures in the revision.

Reviewer: 2

Reviewer Name: Somchit Jaruratanasirikul Institution and Country: Prince of Songkla University
Please state any competing interests or state 'None declared': None

Please leave your comments for the authors below

This is a population-based study of the prevalence of neural tube defects (NTD) in Shanxi County, Northern China, comparing between before folic acid supplementation (2003-2004) and after folic acid supplementation (2011-2016). The results of the study showed a significantly higher prevalence of NTD in females than in males in the time period before folic acid supplementation and a significant overall decrease of the prevalence of NTD after folic acid supplementation, particularly in females.

Comments

1. The Introduction section has 2 pages which is too long. Some parts in paragraphs 2 and 3 can be moved to the Discussion section.
We reorganized introduction and moved several parts from introduction section to discussion section.
2. In the Abstract, the authors stated that "few population-based studies have examined sex difference among infants with NTD". Actually, sex is one of many parameters that are used for comparison in most of the previous studies of the prevalence of NTD; some without sex differences and some with sex differences.
Yes, sex difference of the prevalence of NTDs was mostly compared, while the sentence here we refer was sex difference with the relation with folic acid supplementation was scarce. We revised the expression as we copied as following: "Sex differences in prevalence of neural tube defects (NTDs) have previously been recognized; however the different susceptibility of males and females have not been examined in relation to the effects of folic acid (FA) supplementation."

3. The strength of this study is the measurement of blood folic acid levels. However, the authors should give more details of the blood collection method and the method of folic acid measurement, not just the microbiological assay and a reference. Also, the interassay and intra-assay coefficients of variations should be stated in the method of measurement.
We added the details of blood collection method and method, as well as Intra and inter-assay coefficients of variation in the revision. See second paragraph of MATERIALS AND METHODS.
4. The authors showed the decreased prevalence of NTD and the increased level of folic acid before and after the folic acid supplementation using a histogram. It may be better if the authors showed these results using correlation statistics.
We deleted figures in the revision.
5. Figure 2 shows the blood levels of folic acid comparing participants who took folic acid and those who did not take folic acid, which was already described in the Results section. Hence, Figure 2 should be deleted.
Yes, the information in Figure 2 was described in the last part in Result section. We deleted Figure 2 in the revision.

Reviewer: 3

Reviewer Name: Saeed Dastgiri

Institution and Country: Tabriz University of Medical Sciences, Tabriz, Post Code: 5166615739, Iran
Please state any competing interests or state 'None declared': no competing interests

Please leave your comments for the authors below

This is a study to investigate the association of folic acid supplementation and NTDs (by sex) in northern China. The study is well organized/written. Some points/comments:

(1) How the folic acid supplementation in mothers was determined during the study period? The coverage rate? The accuracy of data? More details are needed.

Description on the folic acid supplementation has been added in the second paragraph in MATERIALS AND METHODS.

(2) The details of the basic characteristics of mothers and neonates are needed (as a separate table).

As it's the population-based surveillance data, individual characteristics was not available in current study.

(3) This is an ecological design in which many surrounding confounding factors might usually affect the data/results. What are the confounders in this study? How investigators have controlled the role of those confounders?

This study was done in the same population. The massive FA supplementation program was the only major factor that may affect the population rate of NTDs during the course of the study. Improvement in living standards and prenatal diagnosis may also impact the rate, but this potential impact should have no difference on males and females. We had acknowledged the limitation in the revision.

(4) The role of the improvements in the diagnosis and surveillance of NTDs over the study period in detection of NTD cases need to be indicated/addressed.

We acknowledge that prenatal diagnosis, especially fetal ultrasound scan, progress with time. In the present study, all NTD cases, regardless of gestational weeks, were included in the surveillance. Pregnancy termination due to improvement in prenatal diagnosis should have impact the differences in rates of males and females. In addition, the improvement in prenatal diagnosis should have no differential impact on males and females. The improvement in prenatal diagnosis and surveillance should have no major impact on the conclusion of the present study.

Reviewer: 4

Reviewer Name: Vijaya Kancherla

Institution and Country: Emory University, USA

Please state any competing interests or state 'None declared': None declared.

Please leave your comments for the authors below

This is an important work and a well-written paper. The authors have very clearly presented the rationale and findings on the impact of folic acid supplementation on overall reduction in NTDs, and sex-specific reductions. I have a few comments for minor revisions and to require some clarifications.

1. There are a few acronyms in the abstract and manuscript that need to be defined at their first use. (e.g., NTD)

We had added the full name of the abbreviate words in the abstract, for example, neural tube defects (NTDs), and acronyms in the manuscript.

2. Refer to studies from the US and other countries with active population-based birth defects surveillance registries to comment on sex difference in prevalence of NTDs (overall and by NTD subtype) during pre- and post-mandatory folic acid fortification periods. There is a reference from Mexico; however, it would be of interest to see what was observed in other countries that have data to compare with the current study.

The published data were focused on countries with folic acid fortification such as U.S., South America, while few study has reported the situation of folic acid supplementation.

3. In the Methods section, clarify how sex-specific prevalence was estimated. Who were included in the denominators? It would also be informative if sex-specific prevalence or proportion of cases were presented by pregnancy outcome (live birth / still birth / elective termination) before and after fortification either in the main manuscript or as a supplemental table.

Thanks for the suggestion. We added all the births by sex as sex-specific denominators.

Unfortunately, we only have the detailed sex-specific information about pregnancy outcome of NTDs, so we couldn't calculate the prevalence by pregnancy outcome without the corresponding denominator.

4. On page 9, first para, authors say that the decrease in prevalence of NTD in China post supplementation is still higher than USA, Canada, Germany, England, and Western Australia. US,

Canada and Western Australia have mandatory fortification with folic acid. Would it be possible that the supplementation program in China is still dependent on compliance of women taking the folic acid pills? That would be different in countries with mandatory fortification, as there is no reliance on personal behaviors to take supplements. Are there any data from China on compliance to folic acid pill intake in the target population? If yes, perhaps you can include it in your discussion to discuss why prevalence in China is higher in spite of the overall drop from the pre-supplementation period.

Yes, the compliance of taking folic acid was a very important factor. We had referred study of rate of compliance in the similar target population in the revision.

5. In Table 1, add columns with total births (indicating denominator values) for males and females (i.e., next to frequency of cases for each sex group)

We added the total births in Table 1.

6. Was unable to read the figures clearly the way it was printed in the PDF. Can increase the size and resolution of the figures.

As the information in Figures was described in the last part in Result section. We deleted Figures in the revision.

VERSION 2 – REVIEW

REVIEWER	Sarah C. Tinker Centers for Disease Control and Prevention, USA
REVIEW RETURNED	30-May-2018

GENERAL COMMENTS	The authors have done an excellent job of responding to my comments, as well as those of the other reviewers. I think there are a few places in the paper that can be made more clear by the author, and there are some remaining issues that need to be addressed. It still appears that no aspect of the original research presented in the paper covers assessment of plasma folate concentrations, and that data presented come exclusively from existing data that have already been published. I highly recommend limiting presentation of these data to the Discussion section. Last sentence of Introduction: The authors have not made a compelling case for how sex differences in the prevalence of NTDs or the relative reduction in NTD prevalence after folic acid supplementation campaigns have any relevance for “personalized prevention recommendations for NTDs”. Given that folic acid was effective in reducing the prevalence of NTDs among both males and females, and no doses were assessed, the recommendation for women to take folic acid supplements before and during pregnancy would be made regardless of the fetuses sex, particularly because the gestational time period of NTD development is long before sex can be determined in most pregnancies. Please remove this statement or provide justification. Otherwise, the statement about helping to better understand NTD development can stand alone. In the abstract it says that all live births, stillbirths, and pregnancy terminations that reached 28 gestational weeks were included, and that all NTDs regardless of gestational age were included. In the first
--

	paragraph of the Introduction it says that all live births of 28 or more complete gestational weeks and all still births of at least 20 weeks' gestational age, and terminations at any gestational age with an NTD were included. These statements are not consistent, and it is not clear which descriptions in the Methods apply to all births included in the analysis versus NTDs. Please harmonize and clarify. Again – it is not clear whether the authors actually conducted these cross-sectional surveys on folic acid supplement use and blood folate concentration, or simply cite data from previous publications. There are no data from these surveys presented in the Results. First sentence of Discussion – it is not a sex “bias” since no estimate is being biased. I believe a more appropriate descriptor would be “changes in the sex ratio for NTD prevalence”, or something similar. The authors need to mention somewhere in the Discussion that the larger decline in NTD prevalence among females is somewhat expected given that it was higher in the pre-supplementation time period and that the fact that the difference between sexes is less pronounced is consistent with prevalences beginning to converge toward the prevalence of NTDs that are not susceptible to folic acid. First paragraph on p. 10 – there is a statement that says the prevalence was higher in 2016 than in 2014, but the prevalence given in the parentheses is identical for both time periods. Second paragraph p. 10 – Is this the description of the data from the 2 surveys? Third paragraph p. 10 – The final two sentences are not clear. During which years was the median 40.0 nmol/L for those who took FA supplements and 13.2 for those who did not?
--	---

REVIEWER	Somchit Jaruratanasirikul Prince of Songkla University, Hat Yai, Songkhla, Thailand
REVIEW RETURNED	22-May-2018
GENERAL COMMENTS	Accept for publication.

VERSION 2 – AUTHOR RESPONSE

Reviewer: 2

Reviewer Name: Somchit Jaruratanasirikul

Institution and Country: Prince of Songkla University, Hat Yai, Songkhla, Thailand

Please state any competing interests or state 'None declared': None declared

Please leave your comments for the authors below

Accept for publication

Thank you!.

Reviewer: 1

Reviewer Name: Sarah C. Tinker

Institution and Country: Centers for Disease Control and Prevention, USA

Please state any competing interests or state 'None declared': None declared

Please leave your comments for the authors below

The authors have done an excellent job of responding to my comments, as well as those of the other reviewers. I think there are a few places in the paper that can be made more clear by the author, and there are some remaining issues that need to be addressed.

It still appears that no aspect of the original research presented in the paper covers assessment of plasma folate concentrations, and that data presented come exclusively from existing data that have already been published. I highly recommend limiting presentation of these data to the Discussion section.

Thank you for the constructive comments. We moved this section to the Discussion section (See fourth and fifth paragraph in the Discussion section).

Last sentence of Introduction: The authors have not made a compelling case for how sex differences in the prevalence of NTDs or the relative reduction in NTD prevalence after folic acid supplementation campaigns have any relevance for "personalized prevention recommendations for NTDs". Given that folic acid was effective in reducing the prevalence of NTDs among both males and females, and no doses were assessed, the recommendation for women to take folic acid supplements before and during pregnancy would be made regardless of the fetuses sex, particularly because the gestational time period of NTD development is long before sex can be determined in most pregnancies. Please remove this statement or provide justification. Otherwise, the statement about helping to better understand NTD development can stand alone.

Thank you for the suggestion. We removed this statement in the revision (See last sentence in the Introduction section).

In the abstract it says that all live births, stillbirths, and pregnancy terminations that reached 28 gestational weeks were included, and that all NTDs regardless of gestational age were included. In the first paragraph of the Introduction it says that all live births of 28 or more complete gestational weeks and all still births of at least 20 weeks' gestational age, and terminations at any gestational age with an NTD were included. These statements are not consistent, and it is not clear which descriptions in the Methods apply to all births included in the analysis versus NTDs. Please harmonize and clarify.

Thank you for the suggestion. The statement in the Method was correct: All live births (28 or more complete gestational weeks), all stillbirths of at least 20 weeks' gestational age, and pregnancy

terminations at any gestational age following the prenatal diagnosis of NTDs were included. We clarified the description in the abstract and harmonized throughout the paper in the revision.

Again – it is not clear whether the authors actually conducted these cross-sectional surveys on folic acid supplement use and blood folate concentration, or simply cite data from previous publications. There are no data from these surveys presented in the Results.

Thank you for the constructive comments, as we replied in the beginning, we moved the information on the two cross-sectional surveys section to the Discussion section (See fourth and fifth paragraph in the Discussion section).

First sentence of Discussion – it is not a sex “bias” since no estimate is being biased. I believe a more appropriate descriptor would be “changes in the sex ratio for NTD prevalence”, or something similar.

Thank you for the suggestion. We revised the description as you suggested “changes in the sex ratio for NTD prevalence”. (See first sentence in the Discussion section).

The authors need to mention somewhere in the Discussion that the larger decline in NTD prevalence among females is somewhat expected given that it was higher in the pre-supplementation time period and that the fact that the difference between sexes is less pronounced is consistent with prevalences beginning to converge toward the prevalence of NTDs that are not susceptible to folic acid.

Thanks for your profound insight. We added this part before the limitation in the discussion.

First paragraph on p. 10 – there is a statement that says the prevalence was higher in 2016 than in 2014, but the prevalence given in the parentheses is identical for both time periods.

Thanks for the suggestion. We are sorry for citing the wrong data, after careful check the original data, we revised in the revision “the prevalence in the same area was higher in 2015-2016 (32.8 per 10,000 births) than in 2014 (31.5 PubMed per 10,000 births).”

Second paragraph p. 10 – Is this the description of the data from the 2 surveys?

Yes, it was from the previous two surveys. We added one sentence to make it clearer “Our previous cross-sectional study showed that the proportion of people taking FA supplements increased from 9.2% in 2002–2004³³ to 66.3% in 2011–2012²¹”

Third paragraph p. 10 – The final two sentences are not clear. During which years was the median 40.0 nmol/L for those who took FA supplements and 13.2 for those who did not?

It referred to 2003–2004, PubMed we added in the revision “Women who took FA supplements had significantly higher folate concentrations (median 41.9 nmol/L) than those who did not (13.2 nmol/L) in 2003–2004.” PubMed ;

As the second and third paragraph in p. 10 were discussing the similar topic, i.e, the proportion of FA supplementation and folate concentration, we hence merged them into one paragraph in the revision (See fifth paragraph in the Discussion section).

VERSION 3 – REVIEW

REVIEWER	Sarah Tinker Centers for Disease Control and Prevention
REVIEW RETURNED	22-Aug-2018
GENERAL COMMENTS	The authors have adequately addressed my remaining concerns.